# Test-Retest Reliability and Walk Score^®^ Neighbourhood Walkability Comparison of an Online Perceived Neighbourhood-Specific Adaptation of the International Physical Activity Questionnaire (IPAQ)

**DOI:** 10.3390/ijerph16111917

**Published:** 2019-05-30

**Authors:** Levi Frehlich, Anita Blackstaffe, Gavin R. McCormack

**Affiliations:** Department of Community Health Sciences, Cumming School of Medicine, University of Calgary, Calgary, AB T2N 4Z6, Canada; ambortni@ucalgary.ca (A.B.); Gavin.McCormack@ucalgary.ca (G.R.M.)

**Keywords:** walkable environment, physical activity, sedentary behaviour, neighbourhood, walkability, active living, survey, questionnaire

## Abstract

There is a growing public health interest in the contributions of the built environment in enabling and supporting physical activity. However, few tools measuring neighbourhood-specific physical activity exist. This study assessed the reliability of an established physical activity tool (International Physical Activity Questionnaire: IPAQ) adapted to capture perceived neighbourhood-specific physical activity (N-IPAQ) administered via the internet and compared N-IPAQ outcomes to differences in neighbourhood Walk Score^®^. A sample of *n* = 261 adults completed an online questionnaire on two occasions at least seven days apart. Questionnaire items captured walking, cycling, moderate-intensity, and vigorous-intensity physical activity, undertaken inside the participant’s perceived neighbourhood in the past week. Intraclass correlations, Spearman’s rank correlation, and Cohen’s Kappa coefficients estimated item test-retest reliability. Regression estimated the associations between self-reported perceived neighbourhood-specific physical activity and Walk Score^®^. With the exception of moderate physical activity duration, participation and duration for all physical activities demonstrated moderate reliability. Transportation walking participation and duration was higher (*p* < 0.05) in more walkable neighbourhoods. The N-IPAQ administered online found differences in neighbourhoods that vary in their walkability. Future studies investigating built environments and self-reported physical activity may consider using the online version of the N-IPAQ.

## 1. Introduction

There is a growing public health interest in the contributions of the built environment in enabling and supporting physical activity. Several systematic reviews provide consistent evidence for an association between neighbourhood built characteristics and physical activity [1,2,3]. While this evidence is encouraging, the accuracy of estimated associations between the built environment and physical activity have been limited by the use of non-context specific physical activity measures; that is, measures of physical activity associated with built environment characteristics have not been specific to the neighbourhood, rather capturing behaviour regardless of the location. There have been previous calls for the development and use of context-specific physical activity measures [3]; however, few context-specific self-report measures of neighbourhood-specific physical activity, in particular, exist [4,5]. Ideally, the use of objective measures of physical activity are preferable; however, these approaches for physical activity measurement are not always feasible, especially in large population-based studies or for research teams with limited financial resources and technical expertise to purchase and use accelerometers and global positioning system (GPS) monitors.

Several studies have captured neighbourhood-specific physical activity via self-report questionnaires [4,5,6,7]. To date, psychometric testing of items capturing self-reported neighbourhood-specific physical activity have included primarily assessments of test-retest reliability [4,7,8]. For instance, the Neighbourhood Physical Activity Questionnaire (NPAQ) that captures transportation and recreational walking and cycling (i.e., traveling to and from work, doing errands, or going from place to place) inside the neighbourhood (within a 15-min walking distance from home) during a usual week has been demonstrated to have adequate reliability in Australian adults [4]. Furthermore, a modified version of the NPAQ has also been found to be reliable in Canadian adults [8]. Physical activity estimates derived from the NPAQ have been associated with neighbourhood built characteristics in Australian [4], Canadian [8], and Chinese [7] populations. However, available options of tools capturing neighbourhood-specific physical activity are limited, and of those that exist, most capture “usual” week behaviour that may overestimate physical activity undertaken in the past week. Although not providing a habitual measure of behaviour, the latter recall type might be more sensitive to immediate changes in the neighbourhood built environment or other neighbourhood-level interventions on physical activity [9]. However, while items for capturing self-reported neighbourhood-specific physical activity in the “last” week have been used in previous built environment–physical activity studies [5,6], none to the best of our knowledge have undergone a thorough assessment of reliability.

Studies using internet-administered questionnaires to capture physical activity are becoming more frequent [10,11,12,13]. Internet questionnaires, despite some limitations, have several advantages over “paper-and-pencil” questionnaires, such as being low cost, allowing the researcher more control over the sequence in which questions and response options are presented, as well as offering skip patterns, allowing the researcher to embed procedures for reducing incomplete data, to instantaneously monitor questionnaire completion and survey participation, and to incorporate visual and audio prompts within the questionnaire to aid comprehension [14]. Internet-based questionnaires are being used more often in population surveys [14,15,16], which has been facilitated by the continued growth in internet access and availability in relation to signal coverage, increased number of internet connected devices or devices offering internet connection, and social and cultural shifts in relation to the perceived importance and expectation of being continually connected to the internet or having internet access [17,18]. It is estimated that globally approximately 52% of households have access to the internet, while in developed countries about 83% have internet access [19]. 

The International Physical Activity Questionnaire (IPAQ) is widely used in physical activity research and has demonstrated reliability and validity [20]. The IPAQ has also been shown to be reliable when used in an online format to measure physical activity in Danish [11] and UK [13] populations. The IPAQ has been adapted to capture perceived neighbourhood-specific physical activity (N-IPAQ) [21,22]. The aim of this study was to adapt, administer, and test the N-IPAQ via the internet in a Canadian population. Specifically, we aimed to assess the reliability of the N-IPAQ and compare differences in self-reported perceived neighbourhood-specific physical activity found from the N-IPAQ by neighbourhood walkability measured with Walk Score^®^. Our goal was to provide researchers interested in capturing neighbourhood-specific physical activity with a reliable online self-report option, capturing past week physical activity undertaken within perceived residential neighbourhoods.

## 2. Materials and Methods 

### 2.1. Study and Sample Design

The sample list for the current methods study was derived from a database of participants from a larger study (i.e., *Pathways to Health)* investigating the relations between neighbourhoods, physical activity, diet, and weight status. The methods for the *Pathways to Health* study have been described in detail elsewhere [23,24]; however, briefly, in April 2014 *n* = 173 established Calgary (Alberta, Canada) neighbourhoods were stratified into 12 strata. Strata were based on the neighbourhood street pattern (grid, warped grid, and curvilinear) and socioeconomic status (advantaged, somewhat advantaged, somewhat disadvantaged, and disadvantaged). Following computer automation, a random sample of *n* = 10,500 households were sent recruitment postcards. A total of *n* = 1023 participants completed a physical activity, health, and demographic questionnaire (PAHDQ). Within the Calgary context, neighbourhoods with grid street patterns were more supportive of walking (high walkability) compared to neighbourhoods with warped-grid (medium walkability) and curvilinear street patterns (low walkability) [25]. Grid street pattern neighbourhoods, typically found centrally in urban areas, have higher land use or destination mix, more street and pedestrian connectivity, and higher population or residential density than those curvilinear street pattern neighbourhoods that are typically found in suburban areas. Land use mix, connectivity, and density are consistent correlates of physical activity [26] and considered to be important built characteristics for determining the walkability of a neighbourhood [27]. Furthermore, walking for transportation has been consistently linked to neighbourhoods with higher population density, distance to non-residential destinations, and proximal non-residential destinations [1]. These relationships have been tested in different countries, such as Canada [28], Australia [29], France [30], Sweden [6], and the United States [31], with studies finding an increased likelihood of transportation walking with increasing access to services and street connectivity. The estimation of neighbourhood socioeconomic status, described elsewhere [23,24], was based on 2006 Canadian census data and included the proportion of those 25–64 years of age who obtained less than a high school diploma; the proportion of single-parent families and the proportion of divorced, separated, or widowed among those ≥15 years of age; the proportion of individuals renting private dwellings and the average value of dwellings; the proportion of unemployment among those ≥25 years of age, and median gross household income. 

The *Pathways to Health* participants also noted their willingness (and contact details) to participate in follow-up research. Thus, in January 2017 we approached all participants who were willing to participate in follow-up research and who did not complete the reliability and validity testing of the paper N-IPAQ (*n* = 515). Of those approached, *n* = 151 did not respond, *n* = 65 refused to participate, *n* = 18 emails returned undeliverable, and *n* = 281 consented to participate in the study. Those consenting were sent instructions and a web-link for completing the online N-IPAQ on two occasions (i.e., Time 1 and Time 2) at least seven days apart. The Time 1 and Time 2 questionnaires were identical with respect to physical activity variables; however, sociodemographic variables were only collected at Time 1. SurveyMonkey^®^ (SurveyMonkey Inc, San Mateo, California, USA) was used for creating and delivering the questionnaire and for collecting responses. Completion of all questions was mandatory in order to progress through the questionnaire (i.e., from page to page) and to submit the questionnaire. Participants who completed both Time 1 and Time 2 questionnaires were entered into a prize draw with a 1 in 40 chance to win a gift card ($50 value). Data collection occurred between January and March 2017. The University of Calgary Conjoint Health Research Ethics Board approved this study (Ethics ID: REB15-2940).

### 2.2. Variables

#### 2.2.1. Neighbourhood Adapted International Physical Activity Questionnaire (N-IPAQ)

We modified items from the IPAQ (long-form) capturing frequency (number of days) and usual minutes (on one of the reported days) of perceived neighbourhood-specific walking for transportation, bicycling for transportation, walking for leisure, moderate-intensity physical activity, and vigorous-intensity physical activity during the last seven days. The original questionnaire item wording was modified to reflect our focus on capturing neighbourhood-specific physical activity by inserting the phrase “…inside your neighbourhood”. For example, “During the last 7 days, on how many days did you walk to go from place to place inside your neighbourhood?”. Within the questionnaire, participants were provided with fixed response options for physical activity frequency (i.e., 0 to 7 days) and duration (i.e., 5 to 480 min/day, in 5-min increments). Participants were instructed to think about activities undertaken inside their residential neighbourhood; however, we did not provide an operational definition for neighbourhood, and instead we permitted the participant to interpret “residential neighbourhood” in their own way. By not placing limits on the neighbourhood geographical area (e.g., 400 m or 5 min walk) or boundary (e.g., administrative boundary), we may have obtained a better understanding on how residents perceive their neighbourhood in terms of a context for physical activity behaviour. Previous research has found a paper administrated version of the N-IPAQ to be reliable [22] and valid compared with accelerometers and GPS monitors [21]. Furthermore, our intention was to go beyond the current assessment of online reliability in future research by comparing self-reported neighbourhood physical activity (using the online N-IPAQ) with context-specific physical activity captured using accelerometers and GPS monitors. The N-IPAQ can be found online under Appendix A.

#### 2.2.2. Sociodemographic Characteristics

Sociodemographic characteristics collected at Time 1 included participants’ sex and age, the number of dependents living at home, the number of dogs living in the household, their access to a motor vehicle for personal use, their access to a bicycle for personal use, and if they had post-secondary school education.

#### 2.2.3. Neighbourhood Walkability

We linked participants’ six-digit residential postal code to Walk Score^®^ to estimate walkability (www.walkscore.com). In the Canadian urban context, a six-digit postal code corresponds to a relatively small geographical unit, which maintains participant confidentiality [32]. Walk Score^®^ has been validated in both the United States [33,34] and Canada [35]. Furthermore, Walk Score^®^ is positively associated with other objective indicators of neighbourhood walkability [36] and walking behaviour [37]. Walk Score^®^ was broken up into tertiles to define low, medium, and high neighbourhood walkability. Walk Score^®^ uses a proprietary algorithm in which amenities located close to home are assigned higher scores and amenities located farther away are assigned a lower score using a distance decay function. Maximum points are received for amenities located within 400 m (a 5 min walk) and a zero score is assigned to amenities located farther than a 30 min walk from home. 

### 2.3. Statistical Analysis

Descriptive statistics including means, standard deviations, and frequencies were calculated for sociodemographic variables for the overall sample and stratified by neighbourhood walkability. We calculated weekly minutes of physical activity by multiplying the reported frequency (in days per week) by the duration of activity (in minutes on a usual day). Participation variables were created by recoding each physical activity into “participation” (at least 1 day/week) versus “no participation” (0 days/week). Similar to previous studies using the IPAQ, to address outliers all variables were truncated at the 99th percentile and weekly physical activity (days × daily duration) was truncated to 1680 min [38,39,40]. 

The reliability of the online N-IPAQ participation variables was assessed using Cohen’s Kappa coefficients (κ) and proportion of overall agreement, while the reliability of the online N-IPAQ continuous variables was assessed using intraclass correlations (ICC) and Spearman’s rank correlations (*ρ*). Furthermore, we estimated ICC and *ρ* among: (1) All participants regardless of participation, and; (2) all participants excluding those who reported zero minutes/week. We used published cut-points for describing our estimates (i.e., ICC, *ρ*, and Kappa correlations: Poor < 0.40, moderate ≥ 0.40 to 0.75, and excellent > 0.75, and proportion of overall agreement ≥75% was considered acceptable) [41]. 

The online N-IPAQ captured perceived neighbourhood-specific differences in physical activity. Highly walkable neighbourhoods have been consistently shown to be associated with more physical activity, especially walking [1,2,30,42]. Notably, several studies have found a higher Walk Score^®^ to be associated with more transportation walking [32,36,37]. We used binary logistic regression to estimate the associations (odds ratios: (OR) and 95% confidence intervals (CI)) between participation in neighbourhood-specific walking for transportation, walking for leisure, bicycling for transport, moderate-intensity and vigorous-intensity physical activity, and neighbourhood walkability. In separate linear regression models, we estimated differences for Walk Score^®^ in relation to weekly minutes of neighbourhood-specific walking for transportation, walking for leisure, bicycling for transport, moderate-intensity and vigorous-intensity physical activity. All models were adjusted for covariates (sex, age, number of dependents, dog ownership, motor vehicle access, bicycle access, and education) and included data from Time 1 only. All statistical analyses were undertaken using STATA version 14.2 (StataCorp, College Station, TX, USA). Statistical significance was set at α ≤ 0.05.

## 3. Results

### 3.1. Sample Characteristics

In total, *n* = 261 participants provided complete data at Time 1 and Time 2 and were included in the analysis. The mean (SD) age of our sample was 53.9 (13.2) years. The sample consisted of 69.4% women, 68.6% with university education, 40.6% dog-owners, 99.7% with access to a motor vehicle, and 82.0% with access to a bicycle. Participants were evenly distributed by neighbourhood walkability (low (35.2%), medium (32.6%), and high (32.2%)). The mean (SD) Walk Score^®^ was 60.8 (14.7); Walk Score^®^ differed significantly for all neighbourhood walkability comparisons (*p* < 0.05 ANOVA: Bonferroni). The average Walk Score^®^ among participants from high walkable neighbourhoods (78.7 (5.5)) was significantly (*p* < 0.05) higher compared with participants from medium (59.8 (5.7)) and low walkable (45.4 (4.8)) neighbourhoods. Participants from low walkable neighbourhoods were on average older than those from medium walkable and high walkable neighbourhoods (*p* < 0.05 ANOVA, Bonferroni: Table 1).

### 3.2. Self-Reported Participation in Perceived Neighbourhood-Specific Physical Activity

The proportion of overall agreement in participation of neighbourhood-specific physical activity between Time 1 and Time 2 ranged from 72.8% for moderate physical activity to 94.6% for bicycling for transport. With the exception of moderate physical activity (72.8%) and walking for recreation (73.2%), the proportion of overall agreement for other physical activities was acceptable (≥75%). Kappa estimated agreement for participation in neighbourhood-specific physical activities was considered moderate (κ = 0.41 to 0.58) (Table 2).

### 3.3. Self-Reported Days and Minutes of Perceived Neighbourhood-Specific Physical Activity

The correlations in self-reported perceived neighbourhood-specific physical activities between Time 1 and Time 2 were moderate for days per week (ICC = 0.50 to 0.66), poor to moderate for usual minutes per day (ICC = 0.37 to 0.57), and moderate for calculated minutes per week (ICC = 0.49 to 0.69) (Table 3). All Spearman’s rank correlations when all participants were included were moderate and ranged from *ρ* = 0.41 for minutes per week of moderate physical activity to *ρ* = 0.75 for computed total minutes active per week (Table 3). After excluding participants who reported zero minutes (no participation), the ICC magnitude increased for all bicycling (days/week: From 0.52 to 0.70, and min/day: From 0.40 to 0.81, min/week: From 0.60 to 0.85) and walking for leisure variables (days/week: From 0.60 to 0.75, min/day: From 0.50 to 0.55, and min/week: From 0.69 to 0.71) (Table 4). After excluding participants who reported no participation, the majority (11/16) of the Spearman’s rank correlations increased in magnitude, with computed total minutes active per week (*ρ* = 0.76) and minutes per week of bicycling for transportation (*ρ* = 0.87) displaying excellent correlations (Table 4).

### 3.4. Relations between Perceived Neighbourhood-Specific Physical Activity and Neighbourhood Built Environment

After adjusting for all covariates, compared with participants in low walkable neighbourhoods, those in high walkable neighbourhoods were more likely to report participation in perceived neighbourhood-specific walking for transportation (OR = 3.02, 95% CI: 1.39 to 6.56) and undertook on average 41.08 (95% CI: 2.87 to 79.30) more minutes per week of walking for transportation. Participants in medium walkable neighbourhoods were more likely to report participation in perceived neighbourhood-specific moderate-intensity physical activity (OR = 2.02, 95% CI: 1.06 to 3.86) than those in low walkable neighbourhoods (Table 5). After adjusting for all covariates, a 1-unit increase in Walk Score^®^ was linearly associated (*p* < 0.05) with a 1.4 min per week increase in perceived neighbourhood-specific walking for transportation (Table 5). Neighbourhood walkability and Walk Score^®^ were not statistically associated with any other physical activity outcomes.

## 4. Discussion

Our findings suggest that IPAQ items, that are adapted to capture self−reported perceived neighbourhood-specific physical activity and are administered via the internet, provide reliable estimates of behaviour. Notably, we also found that self−reported participation and minutes in perceived neighbourhood-specific walking for transportation differed by level of neighbourhood walkability, supporting previous findings showing consistent associations between the built environment and walking [1,5,6]. Although further assessment of the online N-IPAQs measurement validity is needed, our preliminary findings might suggest that this tool is suitable for use in studies investigating relationships between neighbourhood built environments and physical activity, and in particular transportation walking, in adult populations.

With the exception of daily minutes of moderate physical activity (ICC = 0.37), all of our measures had at least moderate test-retest reliability, with the highest test-retest reliability found for total minutes of perceived neighbourhood-specific physical activity during the last week (*ρ* = 0.75). It was not surprising that moderate physical activity outcomes had the lowest reliability statistics as moderate physical activity is often accumulated via many different types of non-walking behaviour that may not be easily recalled. Lower reliability for moderate physical activities captured using the IPAQ has been reported elsewhere [20], including when the IPAQ was administered online [13]. Nonetheless, future research may implement strategies to overcome this limitation, such as researcher-assisted administration, or better clarification and more examples about what constitutes moderate physical activity. The reliability of total physical activity in our study is congruent with previous studies whereby the estimated IPAQ test-retest correlations across 21 studies ranged from 0.46 to 0.96 [20], depending on the lapsed time between administrations (between three and seven days), the data collection mode (self-reported versus telephone interview), the number of administrations (between one and three), and if “usual week” or “last 7 days” was used in the questions. Of the 21 test-retest IPAQ studies, only two closely resembled our data collection procedure (i.e., used the long-form IPAQ, captured physical activity undertaken in the last seven days, and asked participants to complete two administrations); these studies had correlations of 0.72 and 0.79 [20]. However, the days between administrations were “up to” seven days later [20]; as time between our administrations was a “minimum” of seven days, these discrepancies in correlation may be due to participants being better able to recall their responses during the shorter time frame. Furthermore, other research using online administrations of the IPAQ showed a test-retest of total physical activity energy expenditure ICC = 0.58 [11] and total moderate−vigorous physical activity ICC = 0.70 [13]. Moreover, results from research undertaken in Canada produced similar reliability for weekly minutes of moderate physical activity inside the neighbourhood (ICC = 0.38) [8]. These results indicate that an online version of the N-IPAQ is a reliable tool for capturing self-reported perceived neighbourhood-specific physical activity.

Built environment characteristics have been consistently associated with physical activity [2,42] and in particular walking [1,30]. Furthermore, walking for transportation has been consistently linked to neighbourhoods with higher population density, distance to non-residential destinations, and proximal non-residential destinations [1]. These relationships have been tested in different countries, such as Canada [28], Australia [29], France [30], Sweden [6], and the United States [31], with studies finding an increased likelihood of transportation walking with increasing access to services and street connectivity. These findings are consistent with our results; that is, compared to lower walkable neighbourhoods, residents living in high walkable neighbourhoods had higher odds of walking for transportation. Our results also showed that residents from high walkable neighbourhoods undertook an average of 41 more total minutes a week of transportation walking compared to residents in low walkable neighbourhoods. This result is further supported by our finding of a statistically significant linear increase in minutes of transportation walking and Walk Score^®^. An association with Walk Score^®^ and transportation walking was excepted as higher scores are linked with a closer proximity to services and amenities [28,36,37]. Importantly, the online N-IPAQ found associations between the neighbourhood environment and transportation walking in the expected direction. 

The definition of neighbourhood may impact validity as residents and researchers may not have the same operational definition or perception of neighbourhood, leading to the modifiable areal unit problem [43,44]. Adams et al. [45] demonstrated that residents’ neighbourhood perception of a 20-min “time to walk to a destination” produced stronger correlations with objective measures than a 10 or 30-min cut-point did. Furthermore, research using GPS devices found GPS points captured by varying buffers ranged from 28.6 to 97.9%, indicating some buffers may not accurately represent a resident’s exposure to their neighbourhood built environment [43]. In our adaptation of the IPAQ, we did not explicitly define the size of the neighbourhood, thus allowing the tool to capture residents’ physical activity within their perceived residential neighbourhood. A paper administration of the N-IPAQ found that perceived neighbourhood-specific physical activity provided strong agreement with physical activity captured with an accelerometer and GPS at a 400m buffer around the participant’s home [21]; however, similar research is needed to test if these results would be found in the online administration of the N-IPAQ.

Our study has several limitations. Levels of self-reported physical activity are often over-reported [46]. Moreover, there may have been a learning effect of the questionnaire, whereby after the respondent completed Time 1, they were more cognizant to their physical activity for Time 2; thus, this could have attenuated our reliability results. The use of objective measures for capturing physical activity, such as GPS combined with accelerometers, may provide a better option for measuring context-specific physical activity; therefore, as was done with the paper version of the N−IPAQ [21], accelerometers synchronized with GPS monitors may be used in future research with smaller samples as a criterion measure to further validate the online N-IPAQ. We did not find statistical differences in education between levels of neighbourhood walkability; however, we did not measure other proxies of socioeconomic status or variables such as health status; therefore, our comparisons with Walk Score^®^ walkability may have been confounded by other measures that may affect physical activity. Although Walk Score^®^ has been validated against objective built environment characteristics [33,34,35], we did not measure these characteristics directly or measure residents’ perceived neighbourhood environment; therefore, this may have introduced some measurement error. Moreover, Walk Score^®^ measures are for a distinct location; therefore, if our participants perceived their neighbourhood as one outside of the Walk Score^®^ buffer, our estimates would be discordant. Moreover, while our adapted tool was reliable and was sufficiently sensitive to detect changes in physical activity (i.e., transportation walking) based on differences in neighbourhood walkability, other approaches of validation, such as comparing self-reported physical activity captured by this tool against objective measures of physical activity are still required. Compared to the 2014 Calgary Census for our study neighbourhoods, our sample had similar education (68.6 vs. 63.0% in the 2014 Calgary Census obtained postsecondary education), had a higher proportion of women (69.4 vs. 49.7% in the 2014 Calgary Census), was older on average (53.9 vs. 39.0 in the 2014 Calgary Census) [24], and may have been more motivated—recruited from a pool of recent study participants who were willing to be contacted for future research—thus limiting generalizability of our findings. Moreover, participation in bicycling was low in our sample which influenced our estimate ICC for bicycling duration. For example, ICC and *ρ* for usual time spent bicycling for transportation increased from 0.40 to 0.81 and 0.48 to 0.87, respectively, when only looking at participants reporting (*n* = 7) bicycling in the last week. Caution may be needed in using this tool to capture neighbourhood bicycling. More pilot testing of this tool in a sample consisting of a larger number of bicyclists is warranted. Lastly, although the use of an internet-based survey mode is convenient, offers more control over survey response patterns, and is low cost relative to other survey modes [14], some participants may have experienced technical difficulties, whether it be due to software, hardware, or personal computer skills resulting in loss of data and or study drop-out.

## 5. Conclusions

The online N-IPAQ has similar test-retest reliability as the non-modified IPAQ. The online N-IPAQ found differences in neighbourhoods that vary in their walkability. Future studies investigating built environment and physical activity may consider using the online version of the N−IPAQ for capturing perceived neighbourhood-specific physical activity.

## Figures and Tables

**Table 1 ijerph-16-01917-t001:** Sample demographic characteristics by neighbourhood walkability.

Demographic Characteristic	Low Walkable (*n* = 92) Estimate	Medium Walkable (*n* = 85) Estimate	High Walkable (*n* = 84) Estimate	Total (*n* = 261) Estimate
Age in years, mean (SD) *	57.1 (12.5)	52.5 (13.4)	52.0 (13.2)	53.9 (13.2)
Female, *n* (%)	61 (66.3)	63 (74.2)	57 (67.9)	181 (69.4)
Dependents living in the home, *n* (%)				
One or more aged <6 years	8 (8.7)	15 (17.7)	17 (20.2)	40 (15.3)
One or more aged 6–18 years	17 (18.5)	25 (29.4)	19 (22.6)	61 (23.4)
Dogs living in the home, *n* (%)	38 (41.3)	36 (42.4)	32 (38.1)	106 (40.6)
Had access to a motor vehicle for personal use, *n* (%)	92 (100.0)	82 (96.5)	81 (96.4)	255 (97.7)
Had access to a bicycle for personal use, *n* (%)	74 (80.4)	67 (78.8)	73 (86.9)	214 (82.0)
Highest level of education, *n* (%)				
Lower than University	35 (38.0)	22 (25.9)	25 (29.8)	82 (31.4)
University	57 (62.0)	63 (74.1)	59 (70.2)	179 (68.6)

* *p* < 0.05. One-Way ANOVA: Bonferroni (continuous variable), Low walkable older than medium and high walkable neighbourhoods. Chi^2^ (categorical variables).

**Table 2 ijerph-16-01917-t002:** Proportion (%) of Overall Agreement (p_0_) and Kappa (κ) coefficients for self-reported physical activity between Time 1 and Time 2.

Physical Activity	Time 1 % (*n*)	Time 2 % (*n*)	p_0_	κ (95% CI)
Bicycled for transportation in perceived neighbourhood	4.2 (11)	6.5 (17)	94.6	0.47 (0.24 to 0.71) *
Walked for transportation in perceived neighbourhood	73.2 (191)	67.8 (177)	82.4	0.58 (0.47 to 0.69) *
Walked for recreation in perceived neighbourhood	57.9 (151)	54.0 (141)	73.2	0.46 (0.35 to 0.57) *
Vigorous physical activity in perceived neighbourhood	44.1 (115)	41.4 (108)	76.6	0.52 (0.42 to 0.63) *
Moderate physical activity in perceived neighbourhood	35.3 (92)	37.9 (99)	72.8	0.41 (0.30 to 0.53) *

* *p* < 0.05. *n* = 261 completed the Time 1 and Time 2 surveys. Seven days elapsed between the Time 1 and Time 2 survey.

**Table 3 ijerph-16-01917-t003:** Intraclass Correlations (ICC) ^#^ and Spearman’s rank correlation (*ρ*) for self-reported perceived neighbourhood physical activity between Time 1 and Time 2 for all participants (*n* = 261).

Physical Activity Measure	Time 1 Mean (SD), Median	Time 2 Mean (SD), Median	ICC (95% CI)	*ρ* (95% CI)
Bicycling for transportation during the last 7 days (in days)	0.13 (0.73), 0	0.15 (0.70), 0	0.52 (0.43 to 0.60) *	0.50 (0.40 to 0.58) *
Usual time spent bicycling for transportation on one of those days (in minutes)	0.77 (4.64), 0	1.78 (7.78), 0	0.40 (0.29 to 0.49) *	0.48 (0.39 to 0.57) *
***Computed: Total transportation minutes/week by bicycle***	2.38 (15.41), 0	4.27 (22.02), 0	0.60 (0.52 to 0.68) *	0.49 (0.40 to 0.58) *
Walking for transportation during the last 7 days (in days)	2.60 (2.39), 2	2.27 (2.30), 2	0.66 (0.58 to 0.72) *	0.67 (0.60 to 0.73) *
Usual time spent walking for transportation on one of those days (in minutes)	23.72 (23.67), 20	20.90 (21.50), 20	0.57 (0.48 to 0.65) *	0.63 (0.55 to 0.70) *
***Computed: Total transportation minutes/week by walking***	92.84 (131.59), 50	77.13 (109.51), 40	0.64 (0.56 to 0.71) *	0.69 (0.62 to 0.75) *
Walking for leisure during the last 7 days (in days)	2.01 (2.39), 1	1.95 (2.44), 1	0.60 (0.52 to 0.67) *	0.55 (0.46 to 0.63) *
Usual time spent walking for leisure on one of those days (in minutes)	26.36 (33.06), 20	25.23 (31.36), 20	0.50 (0.40 to 0.58) *	0.56 (0.47 to 0.64) *
***Computed: Total minutes/week spent walking for recreation, leisure, or exercise***	95.69 (156.25), 30	93.91 (144.50), 25	0.69 (0.61 to 0.74) *	0.58 (0.49 to 0.66) *
Undertaking vigorous physical activity for leisure during the last 7 days (in days)	1.21 (1.67), 0	1.13 (1.69), 0	0.55 (0.46 to 0.63) *	0.55 (0.46 to 0.63) *
Usual time spent in vigorous physical activity for leisure on one of those days (in minutes)	20.67 (28.63), 0	19.27 (33.32), 0	0.55 (0.46 to 0.63) *	0.58 (0.49 to 0.65) *
***Computed: Total minutes/week spent in vigorous physical activity***	59.18 (98.11), 0	56.74 (137.46), 0	0.49 (0.39 to 0.57) *	0.58 (0.49 to 0.66) *
Undertaking moderate physical activity for leisure during the last 7 days (in days)	0.98 (1.67), 0	1.08 (1.78), 0	0.50 (0.41 to 0.59) *	0.48 (0.38 to 0.56) *
Usual time spent in moderate physical activity for leisure on one of those days (in minutes)	13.60 (21.43), 0	16.49 (25.07), 0	0.37 (0.26 to 0.47) *	0.41 (0.31 to 0.51) *
***Computed: Total minutes/week spent in moderate physical activity***	39.18 (76.43), 0	48.54 (94.02), 0	0.49 (0.39 to 0.57) *	0.47 (0.37 to 0.56) *
***Computed: Total minutes/week active ^***	289.25 (298.65), 210	280.59 (295.80), 210	0.70 (0.63 to 0.76) *	0.75 (0.69 to 0.80) *

^#^ Two-way mixed model. * *p* < 0.05. ^ Sum of: Computed: Total transportation minutes/week by bicycle; Computed: Total transportation minutes/week by walking; Computed: Total minutes/week spent walking for recreation, leisure, or exercise; Computed: Total minutes/week spent in vigorous physical activity, and; Computed: Total minutes/week spent in moderate physical activity. Seven days elapsed between the Time 1 and Time 2 survey. SD: Standard deviation. CI: Confidence interval.

**Table 4 ijerph-16-01917-t004:** Intraclass Correlations (ICC) ^#^ and Spearman’s rank correlation (*ρ*) for self-reported perceived neighbourhood physical activity between Time 1 and Time 2 for participants reporting activity.

Physical Activity Measure	*n*	Time 1 Mean (SD), Median	*n*	Time 2 Mean (SD), Median	*n*	ICC (95% CI)	*n*	*ρ* (95% CI)
Bicycling for transportation during the last 7 days (in days)	11	3.09 (1.92), 2	17	2.29 (1.65), 1	7	0.70 (0.00 to 0.94) *	7	0.74 (−0.03 to 0.96)
Usual time spent bicycling for transportation on one of those days (in minutes)	11	18.18 (14.54), 15	17	27.35 (15.52), 25	7	0.81 (0.25 to 0.97) *	7	0.87 (0.35 to 0.98) *
***Computed: Total transportation minutes/week by bicycle***	11	56.36 (53.16), 40	17	65.59 (60.08), 45	7	0.85 (0.37 to 0.97) *	7	0.75 (−0.01 to 0.96)
Walking for transportation during the last 7 days (in days)	191	3.55 (2.10), 3	177	3.35 (2.05), 3	161	0.60 (0.49 to 0.69) *	161	0.59 (0.48 to 0.68) *
Usual time spent walking for transportation on one of those days (in minutes)	191	32.41 (21.99), 30	177	30.82 (19.38), 30	161	0.44 (0.31 to 0.56) *	161	0.52 (0.39 to 0.62) *
***Computed: Total transportation minutes/week by walking***	191	126.86 (139.12), 75	177	113.73 (116.32), 60	161	0.59 (0.48 to 0.68) *	161	0.62 (0.52 to 0.71) *
Walking for leisure during the last 7 days (in days)	151	3.48 (2.18), 3	141	3.62 (2.24), 3	111	0.75 (0.65 to 0.82) *	111	0.73 (0.63 to 0.81) *
Usual time spent walking for leisure on one of those days (in minutes)	151	45.56 (31.83), 40	141	46.70 (28.57), 40	111	0.55 (0.40 to 0.67) *	111	0.66 (0.54 to 0.75) *
***Computed: Total minutes/week spent walking for recreation, leisure, or exercise***	151	165.40 (175.24), 120	141	173.83 (157.44), 120	111	0.71 (0.61 to 0.79) *	111	0.72 (0.62 to 0.80) *
Undertaking vigorous physical activity for leisure during the last 7 days (in days)	115	2.76 (1.44), 3	108	2.74 (1.57), 2	81	0.49 (0.30 to 0.64) *	81	0.47 (0.28 to 0.62) *
Usual time spent in vigorous physical activity for leisure on one of those days (in minutes)	115	46.91 (25.05), 45	108	46.57 (37.60), 40	81	0.44 (0.24 to 0.60) *	81	0.66 (0.51 to 0.77) *
***Computed: Total minutes/week spent in vigorous physical activity***	115	134.30 (108.50), 100	108	137.13 (186.51), 100	81	0.37 (0.17 to 0.54) *	81	0.49 (0.30 to 0.64) *
Undertaking moderate physical activity for leisure during the last 7 days (in days)	92	2.77 (1.71), 2	99	2.85 (1.82), 2	60	0.45 (0.22 to 0.63) *	60	0.50 (0.29 to 0.67) *
Usual time spent in moderate physical activity for leisure on one of those days (in minutes)	92	38.59 (18.36), 30	99	43.48 (21.95), 40	60	0.34 (0.09 to 0.54) *	60	0.46 (0.23 to 0.64) *
***Computed: Total minutes/week spent in moderate physical activity***	92	111.14 (92.77), 90	99	127.98 (114.80), 90	60	0.46 (0.23 to 0.64) *	60	0.48 (0.26 to 0.65) *
***Computed: Total minutes/week active ^***	235	321.26 (297.96), 230	227	322.62 (295.03), 240	217	0.68 (0.60 to 0.74) *	217	0.76 (0.69 to 0.81) *

^#^ Two-way mixed model. * *p* < 0.05. ^ Sum of: Computed: Total transportation minutes/week by bicycle; Computed: Total transportation minutes/week by walking; Computed: Total minutes/week spent walking for recreation, leisure, or exercise; Computed: Total minutes/week spent in vigorous physical activity, and; Computed: Total minutes/week spent in moderate physical activity. Seven days elapsed between the Time 1 and Time 2 survey. SD: Standard deviation. CI: Confidence interval.

**Table 5 ijerph-16-01917-t005:** Associations between self-reported participation in and duration of perceived neighbourhood-based physical activity and Walk Score^®^ measured neighbourhood walkability at Time 1 only (*n* = 261).

**Adjusted Logistic Regression Odds Ratios (OR) for the Association between Participation and Neighbourhood Walkability**						
**Walkability**	**Cycled for Transportation** **OR (95% CI)**	**Walked for Transportation** **OR (95% CI)**	**Walked for Recreation** **OR (95% CI)**	**Vigorous Physical Activity** **OR (95% CI)**	**Moderate Physical Activity** **OR (95% CI)**	
Low	Reference Group	Reference Group	Reference Group	Reference Group	Reference Group	
Medium	0.69 (0.14 to 3.32)	1.20 (0.61 to 2.34)	1.17 (0.62 to 2.21)	1.42 (0.75 to 2.66)	2.02 (1.06 to 3.86) *	
High	0.87 (0.20 to 3.81)	3.02 (1.39 to 6.56) *	0.78 (0.42 to 1.47)	1.32 (0.70 to 2.49)	1.61 (0.83 to 3.12)	
**Adjusted Linear Regression Unstandardized Regression Coefficients (b) for the Association between Duration and Neighbourhood Walkability**						
**Walkability**	**Min/week Cycling for Transportation** **b (95% CI)**	**Min/week Walking for Transportation** **b (95% CI)**	**Min/week Walking for Recreation** **b (95% CI)**	**Min/week Vigorous Physical Activity** **b (95% CI)**	**Min/week Moderate Physical Activity** **b (95% CI)**	**Min/week Total** **Activity ^** **b (95% CI)**
Low	Reference Group	Reference Group	Reference Group	Reference Group	Reference Group	Reference Group
Medium	−0.76 (−5.44 to 3.92)	7.75 (−30.38 to 45.89)	−6.80 (−52.08 to 38.48)	−8.17 (−37.34 to 20.99)	0.63 (−22.29 to 23.54)	−7.35 (−91.91 to 77.20)
High	−1.20 (−5.89 to 3.50)	41.08 (2.87 to 79.30) *	−14.57 (−59.94 to 30.81)	17.64 (−11.59 to 46.87)	9.10 (−13.87 to 32.06)	52.05 (−32.69 to 136.79)
**Adjusted Linear Regression Unstandardized Regression Coefficients (b) for the Association between Duration and Neighbourhood Walk Score^®^**						
**Walkability**	**Min/week Cycling for Transportation** **b (95% CI)**	**Min/week Walking for Transportation** **b (95% CI)**	**Min/week Walking for Recreation** **b (95% CI)**	**Min/week Vigorous Physical Activity** **b (95% CI)**	**Min/week Moderate Physical Activity** **b (95% CI)**	**Min/week Total** **Activity ^** **b (95% CI)**
Walk Score	−0.03 (−0.17 to 0.10)	1.40 (0.32 to 2.47) *	−0.32 (−1.59 to 0.96)	0.50 (−0.32 to 1.33)	0.12 (−0.53 to 0.76)	1.67 (−0.72 to 4.06)

* *p* < 0.05. ^ Sum of: Total of weekly transportation bicycling, transportation walking, recreation walking, vigorous physical activity, and moderate physical activity minutes. Logistic and linear regression models adjusted for age, sex, presence of dependent children under 6 and 6–18 years, dogs living in the household, access to a motor vehicle, access to a bicycle, and education. (**b**) Unstandardized regression coefficient. OR: Odds ratio. CI: Confidence interval. Min/week: Minutes per week.

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
