# Peer review of "Test-Retest Reliability and Walk Score® Neighbourhood Walkability Comparison of an Online Perceived Neighbourhood-Specific Adaptation of the International Physical Activity Questionnaire (IPAQ)"

_ijerph, 2019, doi:10.3390/ijerph16111917_

Round 1

Reviewer 1 Report

In general, the authors articulate their idea well. However, there are some points needed to clarify as follow:

 Introduction: It is well-written. The rational is well-described.

Method: Line 112-115: the authors described how they recruited participants. This is a convenience sample with a low participation rate. Are there differences in demographic characteristics of those who participated and who did not? This will affect generalization aspects of this study, too.

Line 122: A spelling mistake of “January”

Statistical analysis: Are all the PA measures variables normally distributed? In this case, median and IQR should be reported. Why not report Pearson or Spearman correlation coefficients as well? Why not the Bland-Atman plots be used to measure agreement between 2 methods?

Discussion: Is seasonal difference an issue? This study was conducted during January-March.

The authors did not discuss the role of socioeconomic status (SES) which commonly remains a major confounder.

Also, health status of participants was not discussed given that it might affect their physical activity level and how they record their responses.

It’s not clear how the authors assessed the contruct validity of their findings.

How about the “learning effect” since completing the questionnaire the first time could increase awareness of physical activity, that potentially influences the recall of physical activity, which subsequently affects reliability?

How about the perceived neighborhood safety such as neighborhood crime, etc. that can affect PA?

Author Response

Reviewer 1:

In general, the authors articulate their idea well. However, there are some points needed to clarify as follow:

Introduction: It is well-written. The rational is well-described.

1.       Method: Line 112-115: the authors described how they recruited participants. This is a convenience sample with a low participation rate. Are there differences in demographic characteristics of those who participated and who did not? This will affect generalization aspects of this study, too.

a.        We have added information about the comparability of our sample with the Calgary population in our limitations (Lines 442-445). Note, however, our study was primarily focussed on testing the reliability and validity of our tool, not on generating population representative estimates of physical activity behaviour.

2.       Line 122: A spelling mistake of “January”

a.        Spelling and grammar were assessed throughout.

3.       Statistical analysis: Are all the PA measures variables normally distributed? In this case, median and IQR should be reported. Why not report Pearson or Spearman correlation coefficients as well? Why not the Bland-Atman plots be used to measure agreement between 2 methods?

a.        We used standard and established statistical approaches for estimating reliability. Self-reported physical activity duration is often skewed in that there a lot of zero and low reported minutes. Similar to previous studies, we took this skew into account by undertaking the statistical analysis on: 1) all participants regardless of reported minutes, and; 2) among only those participants who reported more than zero minutes (Lines 227-233). The removal of participants reporting zero minutes, improved the distribution of all measured variables.

Based on the reviewer’s suggestion we have split Table 3 into two tables (now Table 3 and Table 4) to provide additional statistical output (mean, standard deviation, median, and Spearman’s rank correlation) for the reader. We have also updated the manuscript to reflect the addition of these statistical outputs (Lines 271-273, 276-280). Bland-Altman plots were not conducted as we did not have a gold-standard comparator of physical activity.

4.       Discussion: Is seasonal difference an issue? This study was conducted during January-March.

a.        Seasonal differences in physical activity are possible. However, our main interest was in estimating reliability and validity of the physical activity tool. It is possible, we would capture a different level of physical activity had the tool been administered at a different time of year; however, the tool would still have the same reliability and validity properties. We are confident that the season in which data were collected would not affect our reliability or validity estimates of our tool.

5.       The authors did not discuss the role of socioeconomic status (SES) which commonly remains a major confounder.

a.        Our sample size did not allow us to explore invariance of measurement by SES. We have noted this in the limitations section (Lines 430-434).

6.       Also, health status of participants was not discussed given that it might affect their physical activity level and how they record their responses.

a.        We have now added this point as a limitation (Lines 430-434).

7.       It’s not clear how the authors assessed the contruct validity of their findings.

a.        We have removed any mention of construct validity from the manuscript.

8.       How about the “learning effect” since completing the questionnaire the first time could increase awareness of physical activity, that potentially influences the recall of physical activity, which subsequently affects reliability?

a.        We have added this point to our limitations (Lines 424-426).

9.       How about the perceived neighborhood safety such as neighborhood crime, etc. that can affect PA?

a.        We have added this to our limitations (Lines 435-436).

Reviewer 2 Report

Specific comments:

Lines 87-88 – The references 21, 22 are not reader-accessible sources. Please remove or provide a link to these documents that are accessible to any reader. Further, how was random selection of the initial sample done? What was the sampling frame/source list for drawing potential participants? What was the response rate? Need more information on methods of initial sample recruitment.

When were the study participants recruited for this study?

Line 102 -- Would be helpful to keep some of the terminology clear--what is ‘transportation walking’? Is it non-leisure walking, i.e. purposeful, destination oriented walking?

Lines 112-115 – The incremental selection of the sample included in this methods study need to be clarified against the initially available sample. Initially 1023 participants were recruited (at random)—line 87—from these a ‘convenient’ sample of 515 were approached and of these 281 were included. This flow of participants need to be more clearly stated and very importantly, the final sample included (281) need to be compared with the initial randomly selected sample, or with Calgary’s comparable population. The readership needs to know how the selected sample for this paper differs from the comparable adult population of residents in Calgary.

Lines 115 and 125 – This is a paper about N-IPAQ. Need to include this tool in the supplementary information with the paper so readers will be able to see the tool for themselves and evaluate it.

Section titled in line 125 – How did the authors ascertain how the respondents defined their neighbourhoods? Were the respondents asked to indicate which neighbourhoods they lived? How did the authors—as a matter of fact, the respondents—handled a scenario where the respondents definition of neighbourhoods might span and include multiple neighbourhoods, e.g. neighbourhood where they live and neighbourhood where they work.

Related to the point above about neighbourhood definitions, the authors state that respondents were allowed to ‘self-define’ the neighbourhoods in addressing the items in the N-IPAQ related to within/inside neighbourhood physical activity. If so, I suggest the title of the paper and within the paper it needs to be clear that when this study refers to neighbourhood these are variously defined by the respondents who participated in the study. In other words, they are subjective or self-perceived neighbourhoods. This is a very important distinction, or caveat, and it needs to be clearly stated in this paper, starting with the title.

Line 148, section on neighbourhood walkability – The proprietary WalkScore is used to assign neighbourhood walkability. The claim is made that the WalkScore is a validated and appropriate measure to classify the degree of walkability in a given neighbourhood. The WalkScore is a conveniently available, though proprietary, measure that might be adequate when there is no other neighbourhood measures on built environment are available. Those researchers who have used the WalkScore and other research tools that measures the built environment, the WalkScore doesn’t always show the valid or robust measure of walkability that it is sometimes made out to be. There is a further complication in this paper. The definition of neighbourhood vis-à-vis the WalkScore and the N-IPAQ needs to be clarified. That is, the walkability of a neighbourhood (the context) vs self-reported physical activity based on the adapted N-IPAQ (the ‘neighbourhood’ based activity)—how well do the two measures of neighbourhoods correspond? (See statement in lines 176-178, as an example, of the clarification needed on the definition of the neighbourhood used in measuring WalkScore and measuring N-IPAQ.)

Lines 178-179 – Confusion has been introduced in this sentence. The sentence in these lines refer to utilitarian walking, whereas elsewhere, earlier, in this paper the focus on the walking behaviour was on “transportation” walking. First, transportation walking needs to be defined in the paper, and second, if transportation walking is not utilitarian walking, then the reference to utilitarian walking needs to be either removed or qualified.

Lines 179-182 – “If our N-IPAQ provides a valid measure of physical activity, we would therefore expect the captured physical activity to differ according to neighbourhood walkability, including Walk Score® in a direction that is consistent with previous evidence.”

This reads as a proposition, or an hypothesis. If so, it should be moved to a place in the paper so the reader sees it earlier. To have a valid test of this hypothesis one needs to define and know what the neighbourhood is. Without clearly defined (and corresponding) geographies both measures—the N-IPAQ and WalkScore—don’t offer a solid reference or foundation. Would the N-IPAQ physical activity measure correlate with any random measure of context, say for example, neighbourhood garbage bins/receptacles (this measure, as an example, is likely not a good measure of ‘walkability’ but does this measure have the same or different correlation with N-IPAQ?).

Lines 198-202 – I am not sure what this sentence really adds: “The mean (SD) Walk Score® was 60.8 (14.7), Walk Score® differed significantly for all neighbourhood walkability comparisons (p<.05 ANOVA: Bonferroni). Average Walk Score® among participants from high walkable neighbourhoods (78.7 (5.5)), was significantly (p<.05) higher compared with participants from medium (59.8 (5.7)) and low walkable (45.4 (4.8)) neighbourhoods.” Isn’t it obvious that the average WalkScore will be higher among participants in high walkable neighbourhoods? (i.e. Discriminant validity)       

Line 224 – from regression analysis a beta coefficient is reported (41.08) and in this same sentence “more minutes” per week of walking for transportation is mentioned. Does the beta coefficient correspond to more minutes of walking? How many more minutes of walking?

Lines 224-228 – These two sentences back to back—don’t they say the same thing. Remove one sentence.

“Participants in medium walkable neighbourhoods were more likely than those in low walkable neighbourhoods to report participation in neighbourhood specific Moderate intensity physical activity (OR=2.02, 95%CI: 1.06 to 3.86). Participants in medium walkable neighbourhoods reported significantly more participation in moderate physical activity than residents from low walkable neighbourhoods (Table 4).”

Line 241 – Title of Table 4. The mention of the word, “objectively” measured, can lead to some confusion in the reader. The objective measure that is been referred to here is the WalkScore. I suggest removing “objectively” and stating that WalkScore neighbourhood walkability in this title.

Line 313 – “their use in large scale population based studies may not be feasible”. I would suggest altering this declarative statement as there are many population studies around the world that have used GPS-paired accelerometers to measure activity and location.

Lines 315-317 – WalkScore is not developed to capture socioeconomic context of neighbourhoods, nor any other social variables. How is this a limitation of the WalkScore? WalkScore, however, as mentioned above, has its own limitation of being a valid measure of built environment of neighbourhoods or even walkability.        

Author Response

Reviewer 2:

1.       Lines 87-88 – The references 21, 22 are not reader-accessible sources. Please remove or provide a link to these documents that are accessible to any reader. Further, how was random selection of the initial sample done? What was the sampling frame/source list for drawing potential participants? What was the response rate? Need more information on methods of initial sample recruitment.

 a.        Thank you for this comment. We have removed the references that were not accessible and added references for the Pathways to Health study methods that provides a detailed description of the random selection, sampling frame, and response rate. We have also added more detail about the Pathways to Health study in the current manuscript (Lines 114-120).

2.       When were the study participants recruited for this study?

 a.        We have added the month and year of initial recruitment (Line 151).

3.       Line 102 -- Would be helpful to keep some of the terminology clear--what is ‘transportation walking’? Is it non-leisure walking, i.e. purposeful, destination oriented walking?

 a.        Thank you for this comment. We have added more detail about ‘transportation walking’ when it first appears in the manuscript (Lines 66-67).

4.       Lines 112-115 – The incremental selection of the sample included in this methods study need to be clarified against the initially available sample. Initially 1023 participants were recruited (at random)—line 87—from these a ‘convenient’ sample of 515 were approached and of these 281 were included. This flow of participants need to be more clearly stated and very importantly, the final sample included (281) need to be compared with the initial randomly selected sample, or with Calgary’s comparable population. The readership needs to know how the selected sample for this paper differs from the comparable adult population of residents in Calgary.

 a.        We have clarified that we approached all participants who were interested in follow up research (Lines 140-142). Moreover, we describe the comparability of our sample to the Calgary population in our limitations (Lines 442-446).

5.       Lines 115 and 125 – This is a paper about N-IPAQ. Need to include this tool in the supplementary information with the paper so readers will be able to see the tool for themselves and evaluate it.

 a.       We have clarified throughout that this study was testing the online version of the N-IPAQ, and stated that reliability and validity statistics of a paper version of the N-IPAQ have been published elsewhere (Line 98). We have also included the N-IPAQ as Supplementary Material.

6.       Section titled in line 125 – How did the authors ascertain how the respondents defined their neighbourhoods? Were the respondents asked to indicate which neighbourhoods they lived? How did the authors—as a matter of fact, the respondents—handled a scenario where the respondents definition of neighbourhoods might span and include multiple neighbourhoods, e.g. neighbourhood where they live and neighbourhood where they work.

 a.        Thank you for this comment, this is a valid point. In the preamble to the questionnaire we state, “Think about whether your activities were carried out inside your residential neighbourhood.” We have clarified this in the manuscript (Line 190-193). Moreover, based on our experience and pilot testing, adults report “neighbourhood” physical activity based on their residential neighbourhood. This makes sense given evidence that most walking and other physical activities are undertaken close to home (not the workplace). Further, we have also published a paper (Frehlich, L., Friedenreich, C., Nettel-Aguirre, A., Schipperijn, J., & McCormack, G. R. (2018). Using Accelerometer/GPS Data to Validate a Neighborhood-Adapted Version of the International Physical Activity Questionnaire (IPAQ). Journal for the Measurement of Physical Behaviour, 1(4), 181-190. doi:10.1123/jmpb.2018-0016) showing strong agreement between N-IPAQ and objectively-measured physical activity (GPS/Accelerometer) within a buffer of 400m of participants home. This agreement was stronger than for other larger neighbourhood definitions implying that participants are reporting physical activity based on the N-IPAQ that is within their residential neighbourhood. We have noted this in our discussion (Lines 418-422).

7.       Related to the point above about neighbourhood definitions, the authors state that respondents were allowed to ‘self-define’ the neighbourhoods in addressing the items in the N-IPAQ related to within/inside neighbourhood physical activity. If so, I suggest the title of the paper and within the paper it needs to be clear that when this study refers to neighbourhood these are variously defined by the respondents who participated in the study. In other words, they are subjective or self-perceived neighbourhoods. This is a very important distinction, or caveat, and it needs to be clearly stated in this paper, starting with the title.

 a.        This is a valid point. We have updated the manuscript throughout, and the title, to reflect the N-IPAQ captures perceived neighbourhood.

8.       Line 148, section on neighbourhood walkability – The proprietary WalkScore is used to assign neighbourhood walkability. The claim is made that the WalkScore is a validated and appropriate measure to classify the degree of walkability in a given neighbourhood. The WalkScore is a conveniently available, though proprietary, measure that might be adequate when there is no other neighbourhood measures on built environment are available. Those researchers who have used the WalkScore and other research tools that measures the built environment, the WalkScore doesn’t always show the valid or robust measure of walkability that it is sometimes made out to be. There is a further complication in this paper. The definition of neighbourhood vis-à-vis the WalkScore and the N-IPAQ needs to be clarified. That is, the walkability of a neighbourhood (the context) vs self-reported physical activity based on the adapted N-IPAQ (the ‘neighbourhood’ based activity)—how well do the two measures of neighbourhoods correspond? (See statement in lines 176-178, as an example, of the clarification needed on the definition of the neighbourhood used in measuring WalkScore and measuring N-IPAQ.)

 a.        This is a valid point and we hope we clarified this for the reader by addressing that the N-IPAQ is a perceived neighbourhood measure (throughout the manuscript), and that Walk Score® is context specific; therefore, there may be discordance in the measures (Lines 437-439).

9.       Lines 178-179 – Confusion has been introduced in this sentence. The sentence in these lines refer to utilitarian walking, whereas elsewhere, earlier, in this paper the focus on the walking behaviour was on “transportation” walking. First, transportation walking needs to be defined in the paper, and second, if transportation walking is not utilitarian walking, then the reference to utilitarian walking needs to be either removed or qualified.

 a.        We have changed utilitarian to transportation (Line 237).

10.    Lines 179-182 – “If our N-IPAQ provides a valid measure of physical activity, we would therefore expect the captured physical activity to differ according to neighbourhood walkability, including Walk Score® in a direction that is consistent with previous evidence.”

 This reads as a proposition, or an hypothesis. If so, it should be moved to a place in the paper so the reader sees it earlier. To have a valid test of this hypothesis one needs to define and know what the neighbourhood is. Without clearly defined (and corresponding) geographies both measures—the N-IPAQ and WalkScore—don’t offer a solid reference or foundation. Would the N-IPAQ physical activity measure correlate with any random measure of context, say for example, neighbourhood garbage bins/receptacles (this measure, as an example, is likely not a good measure of ‘walkability’ but does this measure have the same or different correlation with N-IPAQ?).

 a.        We have deleted this statement and addressed reference to validity throughout the manuscript. Addressing the mention of validity and mentioning that the Walk Score® and N-IPAQ definition of neighbourhood may be discordant, will hopefully allow the reader to better judge the utility of the Walk Score® comparisons. We have also clarified this in the title of the manuscript to prepare the reader upfront.

11.    Lines 198-202 – I am not sure what this sentence really adds: “The mean (SD) Walk Score® was 60.8 (14.7), Walk Score® differed significantly for all neighbourhood walkability comparisons (p<.05 ANOVA: Bonferroni). Average Walk Score® among participants from high walkable neighbourhoods (78.7 (5.5)), was significantly (p<.05) higher compared with participants from medium (59.8 (5.7)) and low walkable (45.4 (4.8)) neighbourhoods.” Isn’t it obvious that the average WalkScore will be higher among participants in high walkable neighbourhoods? (i.e. Discriminant validity)

 a.        This is a valid point. As we did not directly choose participants from high, medium, and low walkability neighbourhoods, we used ANOVA to be sure we had enough variation in our sample to justify the tertile of groups.       

12.    Line 224 – from regression analysis a beta coefficient is reported (41.08) and in this same sentence “more minutes” per week of walking for transportation is mentioned. Does the beta coefficient correspond to more minutes of walking? How many more minutes of walking?

 a.        The beta coefficient is a difference in minutes per week using low walkable neighbourhood as a reference. We have reworded this sentence to make it clearer (Lines 291-293).

13.    Lines 224-228 – These two sentences back to back—don’t they say the same thing. Remove one sentence.

 “Participants in medium walkable neighbourhoods were more likely than those in low walkable neighbourhoods to report participation in neighbourhood specific Moderate intensity physical activity (OR=2.02, 95%CI: 1.06 to 3.86). Participants in medium walkable neighbourhoods reported significantly more participation in moderate physical activity than residents from low walkable neighbourhoods (Table 4).”

 a.        We have removed the latter sentence.

14.    Line 241 – Title of Table 4. The mention of the word, “objectively” measured, can lead to some confusion in the reader. The objective measure that is been referred to here is the WalkScore. I suggest removing “objectively” and stating that WalkScore neighbourhood walkability in this title.

 a.        We appreciate this suggestion and have changed the title of Table 4 accordingly.

15.    Line 313 – “their use in large scale population based studies may not be feasible”. I would suggest altering this declarative statement as there are many population studies around the world that have used GPS-paired accelerometers to measure activity and location.

 a.        We appreciate this suggestion. We have removed the declaration in the sentence, and focused on the limitation of not using GPS and accelerometer as validation measures (Lines 428-430).

16.    Lines 315-317 – WalkScore is not developed to capture socioeconomic context of neighbourhoods, nor any other social variables. How is this a limitation of the WalkScore? WalkScore, however, as mentioned above, has its own limitation of being a valid measure of built environment of neighbourhoods or even walkability.

 a.        This is a valid point. Although Walk Score® was not designed to measure SES it often tracks with increasing housing costs. Further, higher-walkable areas in Calgary are often where our more affluent populations live. However, we agree that this is not an inherent limitation of Walk Score®; therefore, we have removed this sentence. Moreover, with this sentence removed the following sentence is highlighted, and included limitations to Walk Scores® measurement of the built environment (Lines 437-439).

Round 2

Reviewer 2 Report

Thank you revising the paper as per comments offered on the previous version.